# Tropical methane emissions explain large fraction of recent changes in global atmospheric methane growth rate

Liang Feng[1,2], Paul I. Palmer [1,2✉], Sihong Zhu[2,3], Robert J. Parker [4,5] & Yi Liu[3]

Large variations in the growth of atmospheric methane, a prominent greenhouse gas, are driven by a diverse range of anthropogenic and natural emissions and by loss from oxidation by the hydroxyl radical. We used a decade-long dataset (2010–2019) of satellite observations of methane to show that tropical terrestrial emissions explain more than 80% of the observed changes in the global atmospheric methane growth rate over this period. Using correlative meteorological analyses, we show strong seasonal correlations ($r = 0.6$–$0.8$) between large-scale changes in sea surface temperature over the tropical oceans and regional variations in methane emissions (via changes in rainfall and temperature) over tropical South America and tropical Africa. Existing predictive skill for sea surface temperature variations could therefore be used to help forecast variations in global atmospheric methane.

[1] National Centre for Earth Observation, University of Edinburgh, Edinburgh, UK. [2] School of GeoSciences, University of Edinburgh, Edinburgh, UK. [3] Institute of Atmospheric Physics, Chinese Academy of Sciences, Beijing, China. [4] National Centre for Earth Observation, University of Leicester, Leicester, UK. [5] School of Physics and Astronomy, University of Leicester, Leicester, UK. ✉email: paul.palmer@ed.ac.uk

Atmospheric methane ($CH_4$) absorbs short-wave and long-wave radiation and has a radiative forcing of 0.61 W/m² (see ref. [1]), a third of the value for $CO_2$. After carbon monoxide, it is the largest sink of the hydroxyl radical (OH) and therefore plays a significant role in determining the oxidizing capacity of the global troposphere. Emissions are from wetlands, agriculture (e.g., rice paddies, ruminants, and waste), anthropogenic emissions (e.g., fossil fuel production and consumption), burning of biomass and biofuels, with minor emissions from, for example, geological seepage, termites, inland water, and oceans[2]. Loss processes include oxidation by the hydroxyl radical, microbial consumption in soils, and reaction with chlorine atoms[3]. The resulting lifetime of $CH_4$ is ~9 years.

From 2010 to 2019, the global atmospheric growth rate of $CH_4$, inferred from ground-based measurements, varied from 5 ppb/year to nearly 13 ppb/year, with associated global mean levels increasing by 4% from 1798 to 1866 ppb. The current understanding of these global changes is incomplete, with a growing body of work exploring the role of individual sources and sinks[4]. This period of time coincides with the launch in 2009 of the Japanese Greenhouse gases Observing SATellite (GOSAT,[5]) and in 2017 of the European TROPOMI instrument[6]. These data have progressively provided more insight into regional emissions of $CH_4$ and how they vary in time (e.g., refs. [7–9]).

We report regional a posteriori emission estimates of $CH_4$ across the globe but focus on the role of the tropics and how large climate variations, e.g., the 2014–2016 El Niño[10], have influenced the contrasting variations in emissions across tropical continents. We simultaneously infer regional a posteriori $CH_4$ and $CO_2$ flux estimates[11] directly from GOSAT $XCH_4:XCO_2$ retrievals[12] and in situ $CH_4$ and $CO_2$ data using an ensemble Kalman filter[13,14] and the GEOS-Chem global 3D atmospheric chemistry model, mostly driven by a priori flux inventory estimates ("Methods", see ref. [15]). Our direct use of $XCH_4:XCO_2$ retrievals minimizes error associated with assuming a model $XCO_2$ distribution to estimate $XCH_4$[11]. To study the role of hydrology on regional $CH_4$ emissions across the tropics, we use (1) liquid water equivalent depth (LWE) data from the NASA/DLR Gravity Recovery and Climate Experiment (GRACE), which allows us to study the relationship between $CH_4$ emissions and water table height[16]; (2) the CMAP[17] precipitation dataset inferred from satellite and in situ measurements; and (3) MERRA2 reanalyzed meteorological fields that describe surface temperature and soil moisture ("Methods") from NASA GSFC[18]. To study the relationship between sea-surface temperature (SST) and $CH_4$ emissions, via precipitation distributions, we use the NOAA OISST v2 optimized record of SST inferred from in situ and satellite measurements ("Methods"). We evaluate our model of atmospheric $CH_4$ using a sparse network of ground-based remote sensing measurements ("Methods",[19]).

Our analysis of GOSAT $CH_4$ column data from 2010 to 2019 shows large-scale changes in tropical $CH_4$ emissions that explain more than 80% of the observed global atmospheric growth rate. Over this decadal period, we find that tropical Africa plays the largest role in determining the variation of tropical emissions, followed by tropical South America and India. We find that emissions from mainland and maritime (island nations) of Southeast Asia have reduced over our study period, driven by reduced rainfall. Contrary to a previous study we find no evidence of an upward trend in Indian emissions early in the study period, instead our analysis shows large year-to-year variations that peak during the 2014–2016 El Niño and again during 2017 and 2019. We find that we can explain a significant fraction of changes in $CH_4$ emissions over tropical South America and tropical Africa by large-scale changes in tropical SSTs characterized by indices that describe El Niño and the Indian Ocean Dipole, respectively. We propose that our analysis over tropical Africa represents a

first step towards understanding a new positive climate feedback in the Earth system. This argument is based on (1) our analysis on the strength of the correlation between the Indian Ocean Dipole and $CH_4$ emissions over East Africa (via rainfall variations during the short rain season, October–December); and (2) previous studies that link a warming climate to increases in the magnitude and variation of the IOD (and between the strength of the IOD and rainfall over East Africa).

## Results

**Contribution of the tropics to global methane budget.** Generally, we find that global mean a posteriori emissions of $CH_4$ (Table 1) are higher than a priori values by typically 1–5% (5–30 Tg/year) after 2013, and consistent with reported atmospheric growth rates (Fig. 1, https://www.esrl.noaa.gov/gmd/ccgg/trends_ch4/). Global net mean values are driven by anthropogenic (e.g., coal mining over North China,[20,21]) and biogenic (e.g., tropical wetlands) hotspots (Fig. 1). A posteriori tropical terrestrial $CH_4$ emission estimates typically represent ~60% of the global annual mean total, an increase of 1–10% compared to a priori estimates, and describe 84% of the a posteriori variation in the annual mean growth rate. Figure 1 shows that variations in global a posteriori emissions reflect changes occurring over the tropics, with the largest increases during the 2014–2016 El Niño and in 2018–2019. In contrast, there is a near-steady increase in $CH_4$ emissions from the extratropics over the first six years of our 10-year study period, driven by anthropogenic emissions. We also find that a posteriori emissions have a larger seasonal amplitude than a priori values, consistent with a larger role for one or more seasonal sources (Supplementary Fig. 2). The corresponding global mean a posteriori flux estimates for $CO_2$ (Supplementary Table 1) are consistent with global mean atmospheric growth rates inferred from in situ data (https://www.esrl.noaa.gov/gmd/ccgg/trends_co2/).

**Variation of methane emissions from tropical continents.** Figure 2a and Supplementary Fig. 4a show the geographical regions where $CH_4$ emissions have changed the most over our study period. The 2010–2019 annual mean values (denoted by dashed line in Fig. 2b) for tropical South America, tropical Africa, Tropical Southeast Asia, and India are 80 Tg/year, 66 Tg/year, 54 Tg/year, and 32 Tg/year, respectively, compared to the a priori values of 72 Tg/year, 63 Tg/year, 50 Tg/year, and 29 Tg/year.

Figure 2b illustrates the contrasting temporal emission distributions across the tropics from 2010 to 2019. Even across

**Table 1 Annual global and tropical terrestrial net a priori and a posteriori $CH_4$ fluxes (Tg$CH_4$/year) from 2010 to 2019, inclusively.**

| Year | Global $CH_4$ emissions (Tg$CH_4$/year) | | Tropical land $CH_4$ emissions (Tg$CH_4$/year) | |
|---|---|---|---|---|
| | A priori | A posteriori | A priori | A posteriori |
| 2010 | 567.7 ± 33.6 | 566.1 ± 10.6 | 350.0 ± 18.7 | 352.6 ± 6.7 |
| 2011 | 555.9 ± 33.5 | 557.3 ± 10.3 | 341.5 ± 18.3 | 354.7 ± 6.5 |
| 2012 | 558.0 ± 33.6 | 554.8 ± 10.2 | 339.4 ± 18.4 | 347.0 ± 6.5 |
| 2013 | 557.9 ± 33.6 | 564.4 ± 10.2 | 341.5 ± 18.4 | 356.0 ± 6.6 |
| 2014 | 558.9 ± 33.6 | 574.6 ± 10.2 | 342.4 ± 18.4 | 363.5 ± 6.5 |
| 2015 | 559.6 ± 33.6 | 586.4 ± 10.3 | 342.5 ± 18.2 | 375.4 ± 6.6 |
| 2016 | 549.4 ± 33.6 | 576.9 ± 10.1 | 339.8 ± 18.4 | 360.0 ± 6.6 |
| 2017 | 552.0 ± 33.6 | 585.0 ± 10.1 | 340.4 ± 18.4 | 375.1 ± 6.6 |
| 2018 | 545.6 ± 33.6 | 582.6 ± 10.3 | 340.7 ± 18.4 | 380.5 ± 6.6 |
| 2019 | 552.3 ± 33.6 | 584.6 ± 10.0 | 346.3 ± 18.6 | 377.1 ± 6.6 |

Uncertainties denote the 1−$\sigma$ value.

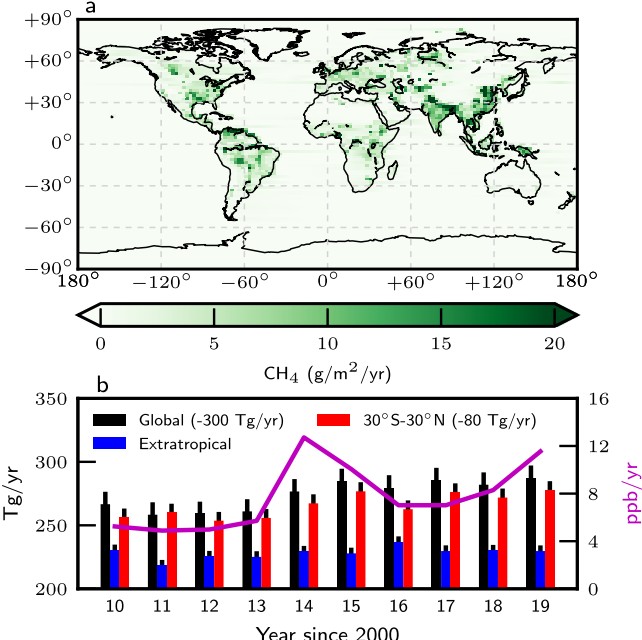

**Fig. 1 Distribution and year-to-year variations of CH₄ emissions. a** Mean annual a posteriori CH₄ emissions (Tg/year) 2010–2019, inclusively, and **b** the corresponding annual mean emissions for the globe (minus 300 Tg) further separated into the tropics (minus 80 Tg) and the extratropics. The thinner vertical lines in **b** denote the 1−σ values about the annual mean values. The purple line denotes the atmospheric growth rate (ppb/year) from the NOAA ground-based network (https://www.esrl.noaa.gov/gmd/ccgg/trends_ch4/).

continents, there are significant variations that can be explained partly by differences in surface temperature and hydrology across the tropics (Supplementary Fig. 4). During our study period, tropical Africa generally plays the largest role in variations of a posteriori tropical emissions of CH₄, followed by tropical South America and India (Fig. 2c).

Emissions from the mainland and maritime tropical Southeast Asia have significantly decreased from 2010 to 2019, driven primarily by reduced rainfall. We find that increased fire emissions due to longer and more frequent droughts are insufficient to balance the inferred long-term downward trend. We find no evidence of an upward trend between 2010 and 2014 in our a posteriori emissions from the Indian subcontinent[21], instead showing large year-to-year variations that peak during the 2014–2016 El Niño and again during 2017 and 2019 beyond an initial period when emissions were comparatively invariant[22].

**Meteorological drivers of tropical methane emissions.** Figure 3, Table 2, and Supplementary Table 2 present a more general examination of the relationship between variations of SST, rainfall, and CH₄ fluxes over Northeast (mostly Columbia, Venezuela, Guyana, and North Brazil) and Southwest (mostly Peru and West Brazil) tropical South America, regions determined by a Thiel-Sen slope analysis of a posteriori CH₄ fluxes (Supplementary Fig. 4). Rainfall variations over Northeast (Fig. 3a) and Southwest (Fig. 3b) tropical South America show distinctly different relationships with SST variations. Rainfall over the Northeast has generally reduced relative to climatology (2000–2019) with large year-to-year variation, associated with positive correlation with changes in SST over the tropical North Atlantic and strong correlation with changes in SST over the eastern equatorial Pacific. Conversely, rainfall over the Southwest during our study period has remained consistently higher than climatological values,

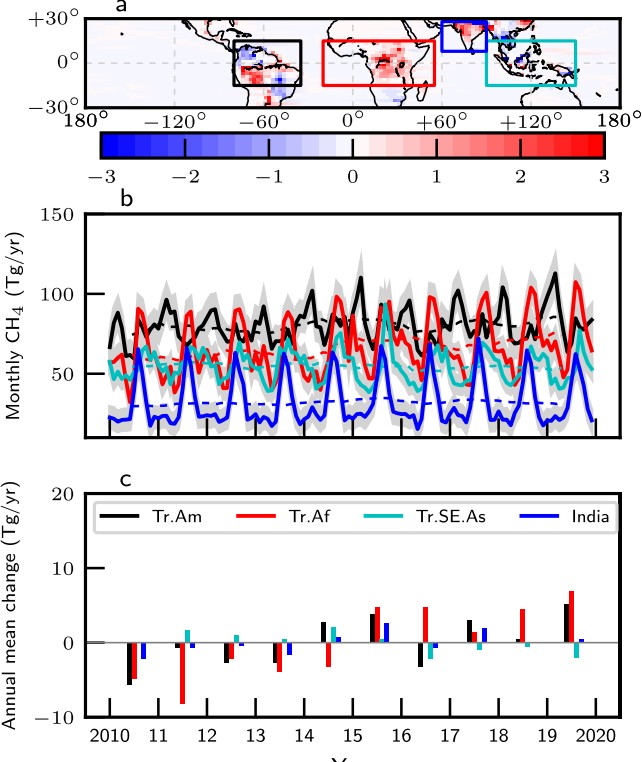

**Fig. 2 Changes in tropical CH₄ emissions. a** The 2015–2019 annual means minus the 2010–2014 annual means of a posteriori tropical CH₄ emissions (Tg/year); **b** corresponding monthly CH₄ fluxes from tropical South America (black line), tropical Africa (red line), India (blue line), and tropical maritime Southeast Asia (cyan line); **c** corresponding annual mean a posteriori CH₄ fluxes relative to their 10-year (2010–2019) means. Shaded envelopes in (**b**) denote the 1 – σ uncertainty. Dashed lines in (**b**) denote the 2010–2019 12-month rolling mean values for each region.

associated with a strong negative correlation with SST variations over tropical North Atlantic. These contrasting signals are reflected in the opposite trends for GRACE LWE over these two regions (Supplementary Fig. 4). Larger-scale dipole structures in these correlations with SSTs are indicative of changes in pan-tropical atmospheric circulation, e.g., Walker circulation[23]. Over our study regions in tropical South America, where CH₄ emissions have changed the most over our 2010–2019 study period (Supplementary Fig. 4), we had anticipated that the Niño 3.4 SST index to provide the best description of variations in CH₄ fluxes (via variations in rainfall, Supplementary Fig. 5), but it is generally outperformed by changes in SST gradients between the tropical Pacific (120°W–90°W, 5°N–20°N) and Atlantic (50°W–30°W, 5°N–20°N) Oceans (Table 2). We find with varying seasonal importance, strong correlations between regional a posteriori CH₄ fluxes (Table 2) and what we expect to be the predominant driving factors of local rainfall and surface temperature (Supplementary Table 3). Over Northeast and Southwest tropical South America, our analysis suggests that changes in rainfall play a dominant role in describing changes in regional CH₄ fluxes, with changes in temperature only playing a small but still significant role (Supplementary Table 3). The r.h.s. panels of Fig. 3 show the result of linear and quadratic regression models that use 3-month moving means of anomalies for rainfall, surface temperature, and soil moisture ("Methods") to describe a posteriori CH₄ flux anomalies over Northeast and Southwest tropical South America inferred from GOSAT. These regression models are highly correlated with the three-month moving mean of a

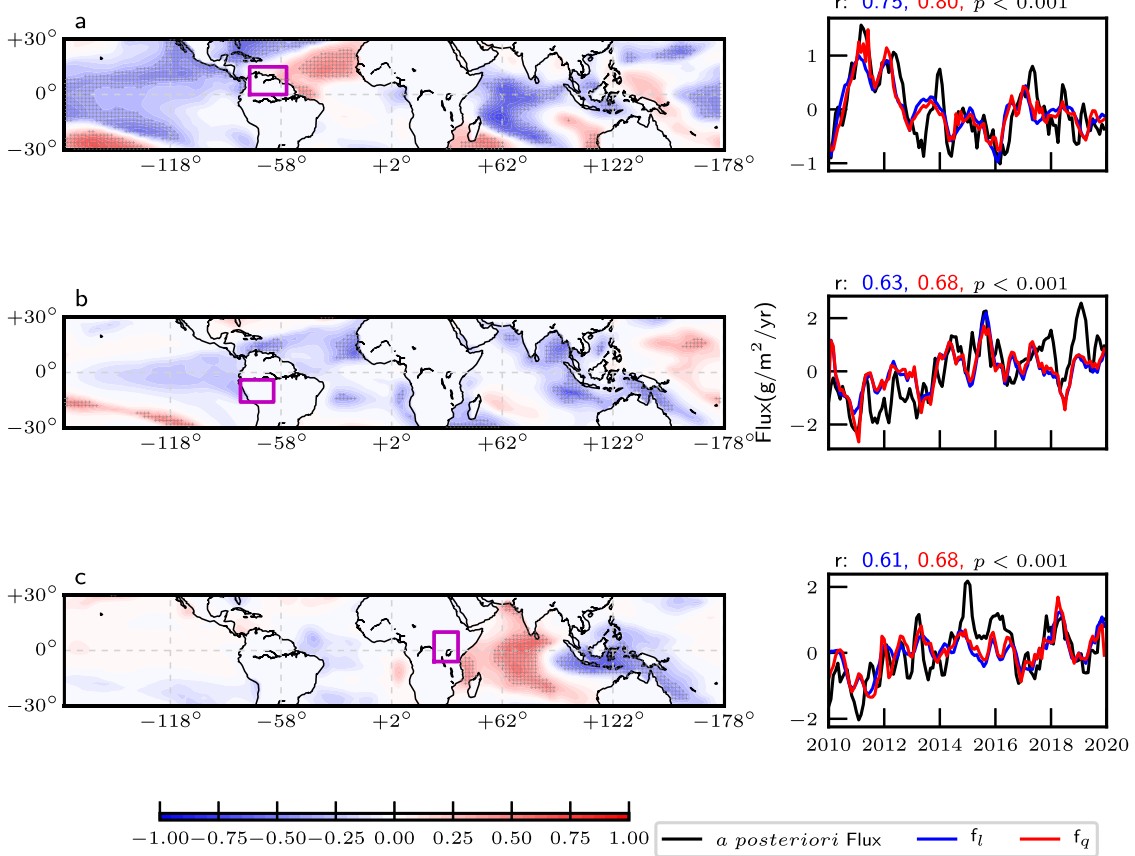

**Fig. 3 Correlations between meteorological analyses and variations in CH$_4$ emissions.** (left) spatial distribution of Pearson correlations ($R^2$, unitless) between monthly values of the NOAA OISST anomaly and CMAP rainfall anomaly and (right) comparison of a posteriori CH$_4$ fluxes (g/m$^2$/year, red lines) with linear (f$_l$) and quadratic (f$_q$) regression models fitted to the 3-month moving means of CMAP rainfall, MERRA2 surface temperature, and MERRA2 soil moisture anomalies (2010–2019) over (**a**) NE tropical South America and (**b**) SW tropical South America during the wet season (DJFM); and over **c** East Africa during the short rains (OND). Purple rectangles in l.h.s. panels denote the geographical regions over which SST and rainfall are correlated. Shaded regions with crosses denote statistical significant values ($P < 0.1$). Blue and red numbers atop of each r.h.s. panel denote the correlation $r$ between the regional a posteriori CH$_4$ flux and the linear and cubic regression models, respectively.

**Table 2 Seasonal mean correlations between (upper table) a posteriori CH$_4$ emission anomalies over NE and SW tropical South America (Fig. 3) and tropical Pacific-Atlantic dipole (PAD) SST anomalies, and (lower table) a posteriori CH$_4$ emission anomalies over tropical Africa (Fig. 3) and Indian Ocean dipole (IOD) SST anomalies.**

|                        | Wet season              | Dry season              |
| ---------------------- | ----------------------- | ----------------------- |
|                        | $n, r, p$ <br> *DJFM*   | $n, r, p$ <br> *JASO*   |
| NE tropical S. America | 10, −0.5, 0.2           | 10, −0.9, <0.001        |
| SW tropical S. America | 10, 0.8, 0.008          | 10, 0.9, 0.02           |
|                        | **Long rains**          | **Short rains**         |
|                        | $n, r, p$ <br> *MAM*    | $n, r, p$ <br> *OND*    |
| East Africa            | 10, −0.3, 0.4           | 10, 0.7, 0.01           |

Variables $n$, $r$, and $p$ denote the number of data points, Pearson correlation coefficient, and the two-tailed $P$ value.

posteriori CH$_4$ flux anomalies ($r = 0.6$–$0.8$), describing 41–65% of CH$_4$ flux variations from 2010 to 2019, with rainfall regression coefficients playing more of a role during wet seasons. We acknowledge that anomalies of rainfall and temperature are not independent of soil moisture anomalies; however, we find that

removing soil moisture from these regression models reduces the correlation with CH$_4$ flux anomalies by only 0.05 ($\simeq$10%).

Figure 3c and Table 2 also show a similar analysis for tropical East Africa. Over this region, we use SST anomalies described by the Indian Ocean dipole (IOD), which has been shown previously to play a significant role in modulating rainfall variations during short rain season (October–December) over East Africa ([24]–[26] and Supplementary Fig. 5). Variations in the short rains determined by the CMAP dataset show a strong positive correlation with SST anomalies over the western Indian Ocean, and a strong negative correlation with SST anomalies over the eastern Indian Ocean, consistent with previous studies using different rainfall datasets (e.g.,[26]). We find that rainfall variations play the dominant role in describing changes in our a posteriori CH$_4$ fluxes (Supplementary Table 3), with temperature variations playing a small but significant role. The resulting linear and quadratic regression models explain up to 62% of monthly a posteriori CH$_4$ flux variations over our study period (Fig. 3c).

## Discussion

Our calculations show that the SST anomalies over tropical oceans have skill in describing variations in a posteriori CH$_4$ flux variations over tropical South America and tropical East Africa, via changes in rainfall (e.g.,[26,27]). Given the global importance of tropical CH$_4$ emissions, these results open up a compelling new

avenue to explore how SST forecasts can be used to help describe future variations in the atmospheric growth of $CH_4$.

Our analysis over tropical Africa, in particular, represents a first step towards understanding a new positive climate feedback in the Earth system. Previous studies have reported relationships between a warming climate due to rising levels of atmospheric greenhouse gases and increases in the magnitude and variations of the IOD[28], and between the strength of the IOD and rainfall over East Africa[29] and, by extension via this study, wetland emissions of $CH_4$. Future changes in the IOD will also impact the large-scale fires over maritime Southeast Asia, where there is a large reservoir of carbon-rich peat, and over Australia. The situation over tropical South America is more complicated with future Atlantic–Pacific SST patterns resulting in regional patterns of anomalous positive and negative rainfall trends over the Amazon basin[30] so that the net regional effect on wetland emissions of $CH_4$ is uncertain.

## Methods

**Satellite proxy retrievals of XCH₄:XCO₂.** We use version 9 of the proxy GOSAT $XCH_4:XCO_2$ retrievals from the University of Leicester[12], including both nadir observations over land and glint observations over the ocean. Analyses have shown that these retrievals have a bias of 0.2%, with a single-sounding precision of ~0.72%[12,31,32]. We globally remove a slightly larger 0.3% bias from the GOSAT proxy data to improve the comparison with independent in situ observations. We find using sensitivity tests (not shown) that uniformly adding/subtracting a bias of this magnitude does not significantly change our results or our conclusions. We also assume that each single GOSAT proxy $XCH_4:XCO_2$ ratio retrieval has an uncertainty of 1.2% to account for possible model errors, including the errors in atmospheric chemistry and transport.

We use these ratios directly, sidestepping any bias introduced by scaling with model or, less often, observed $XCO_2$ values[11]. We achieve this by also ingesting simultaneously $CO_2$ and $CH_4$ mole fraction observations at surface-based sites, which help anchor the GOSAT ratio observations[11]. Supplementary Fig. 1 shows the sites we use from the NOAA observation network[33–36]. We assume uncertainties of 0.5 ppm and 8 ppb for these in situ observations of $CO_2$ and $CH_4$, respectively.

Following previous work[15], we assume a model error of 1.5 ppm and 12 ppb for $CO_2$ and $CH_4$, respectively. We adopt a larger percentage for the $CH_4$ model error to account for difficulties in modeling chemical sinks of $CH_4$ in atmosphere[11,37].

As part of a sensitivity calculation, we also use full-physics GOSAT $XCO_2$ retrievals that represent additional observation constraints on $CO_2$ fluxes. These data are particularly useful over the terrestrial tropics where the in situ data coverage is sparse[38]. We find that using NASA ACOS GOSAT $XCO_2$ retrievals (version 9) results in higher tropical a posteriori $CO_2$ emissions than our control run that uses only GOSAT $XCH_4:XCO_2$ ratio data and in situ $CO_2$ and $CH_4$ data. This subsequently leads to higher tropical $CH_4$ emissions when we use that data together with $XCH_4:XCO_2$ data. We find that the corresponding trends and correlations with SSTs are similar to the control experiment described in the main text.

**Total carbon column observing network.** TCCON is a global network of ground-based Fourier transform spectrometer (FTS) instruments that measure, among other compounds, the total atmospheric columns of $CO_2$ and $CH_4$[19]. Here, we use bias-corrected TCCON $XCO_2$ and $XCH_4$ data to evaluate our a posteriori $CO_2$ and $CH_4$ flux estimates (Supplementary Materials, Supplementary Fig. 3). We use data from all sites from the GGG2014 public release of the TCCON dataset from 2010 to 2019, including updates until August 2020[39,40].

For a comprehensive description of the network and the available data from each TCCON site, we refer the reader to the TCCON project page. Here, we use a subset of available TCCON data[41–74].

**Gravity recovery and climate experiment.** The GRACE space mission was jointly developed by NASA and DLR (German Space Agency) and launched into space in 2002. It measured temporal variations of the Earth's gravity field by tracking through a K-band ranging (KBR) system, the inter-satellite range and range rate between two coplanar, low altitude satellites (GRACE A and B)[75]. The GRACE Science Data System uses measured inter-satellite range and range rate data, along with ancillary data, to estimate monthly (or sub-monthly) time series of global Earth's gravity fields[76,77]. Here, we use the NASA GRCTellus GRACE land product (RL05) for monthly total water storage (liquid water equivalent depth) at $1° \times 1°$ global grids from January 2003 through March 2017[16] (http://grace.jpl.nasa.gov/).

**NASA meteorological reanalyses.** We use surface temperature (TS) and soil moisture (ground wetness, GWET) datasets from the Modern-Era Retrospective Analysis for Research and Applications, version 2 (MERRA2) developed by the Global Modeling and Assimilation Office, NASA Goddard Space Flight Center, to study environmental changes from 2010 to 2019. We refer the reader to refs. [78,79] for a description and an assessment of these datasets.

**Precipitation data.** We use the NOAA CMAP (CPC Merged Analysis of Precipitation) precipitation data and the NOAA OISST v2 sea-surface temperature data together to determine the trends of tropical rainfalls from 2000, as well as their correlation with SST anomalies.

CMAP is a long-term global rainfall dataset[17] that provides near-global coverage at a spatial resolution of $2.5° \times 2.5°$, from 1979 to near-present. The CMAP precipitation rates are obtained from five kinds of satellite data and rain gauge data via a two-stage procedure[17]. First, the random error is reduced by linearly combining the satellite data using the maximum likelihood method. The outputs are used to define the "shape" of precipitation, and their amplitudes are constrained in the second stage by the rain gauge data. We use the enhanced CMAP dataset for which NCEP/NCAR reanalyses are used to fill data gaps; the reader is referred to https://psl.noaa.gov/data/gridded/data.cmap.html#detail for more details. Over tropical lands and oceans, CMAP is similar to another blended rainfall dataset GPCP (The Global Precipitation Climatology Project: https://psl.noaa.gov/data/gridded/data.gpcp.html). Over Africa, we find the long-term trend of CMAP data and the correlations with SST anomalies are similar to the Africa-focused TAMSAT dataset (Tropical Applications of Meteorology using SATellite data and ground-based observations: http://www.tamsat.org.uk/), but are different from CHIRPS (Climate Hazards Group InfraRed Precipitation with Station data, https://www.chc.ucsb.edu/data/chirps#, not shown). These differences reflect the difficulties associated with combining long-term rainfall datasets over tropical regions.

**Sea-surface temperatures.** We use high-resolution optimum interpolation (OI) sea-surface temperature (SST) analyses produced by the National Oceanic and Atmospheric Administration (NOAA) using both in situ and satellite data, (https://psl.noaa.gov) covering nearly 40 years from 1981 to the present[80]. The in situ SST data are determined from observations from ships and buoys. Since 1981, Advanced Very High-Resolution Radiometer satellite retrievals dramatically improved the data coverage by in situ observations. The weekly SST global map is generated by using an optimum interpolation (OI) algorithm to determine increments to first guess (previous week's analysis) according to nearby observations weighted by their distance and error covariances. The comparison with other SST products shows a mean difference of ~0.05 °C on decadal scales[81]. We use the global monthly mean SST product gridded at $1° \times 1°$.

**Atmospheric transport models and inverse methods.** We use an ensemble Kalman Filter (EnKF) framework[11,15] to estimate simultaneously $CO_2$ and $CH_4$ fluxes from in situ and satellite measurements of the atmospheric $CO_2$ and $CH_4$ from 2009 to 2019, inclusively.

Our state vector includes monthly scaling factors for 487 regional pulse-like basis functions (Supplementary Fig. 1) that describe $CO_2$ and $CH_4$ fluxes, including 476 land regions and 11 oceanic regions. We define these our land sub-regions by dividing the 11 TransCom–3[82] land regions into 42 nearly equal sub-regions, with the exception for temperate Eurasia that has been divided into 56 sub-regions due to its large landmass. We use the 11 oceanic regions defined by TransCom–3 experiment.

We assume the a posteriori $CH_4$ or $CO_2$ flux estimate takes the form[15]:

$$f_p^g(x,t) = f_0^g(x,t) + \sum_i c_i^g BF_i^g(x,t), \qquad (1)$$

where $g$ denotes the atmospheric concentration of $CH_4$ or $CO_2$ and $f_0^g(x,t)$ describes their a priori flux inventories, respectively. The pulse-like basis functions $BF_i^g(x,t)$ represent the sum of different source sectors, which we use to represent their overall spatial pattern for each month over each sub-region. As a result, we estimate a total of 128,568 (i.e., 2($CH_4$ or $CO_2$) × 132 (months) × 487 (sub-regions)) coefficients, by optimally fitting model concentrations with observations[15]:

$$\mathbf{c}_a = \mathbf{c}_f + \mathbf{K}[\mathbf{y} - H(\mathbf{c}_f)], \qquad (2)$$

where $\mathbf{c}_a$ and $\mathbf{c}_f$ denote the a priori and a posteriori state vectors, respectively; $\mathbf{y}$ denotes the $CO_2$ and $CH_4$ observations; and $H$ describes the observation operator that relates surface fluxes (i.e., the coefficients) to the observations. For $H$, we use v12.5 of the GEOS-Chem global 3D chemistry transport model[15] that is subsequently sampled at the time and location of each observation and convolved with scene-dependent averaging kernels.

In our EnKF framework, we introduce a flux perturbation (coefficients) ensemble $\Delta\mathbf{C}$ to represent the a priori error covariance and calculate Kalman gain

matrix **K** in Eq. (2) by using

$$\mathbf{K} = \Delta\mathbf{C}\Delta\mathbf{Y}^T \left[\Delta\mathbf{Y}\Delta\mathbf{Y}^T + \mathbf{R}^{-1}\right]^{-1}, \quad (3)$$

where **R** is the observation error covariance, and $\Delta\mathbf{Y} = H(\Delta\mathbf{C})$ represents the projection of the flux perturbation ensemble to observation space. For the experiments reported here, we run GEOS-Chem at a horizontal resolution of 2° (latitude) × 2.5° (longitude), driven by the MERRA2 meteorological reanalyses from the Global Modeling and Assimilation Office Global Circulation Model based at NASA Goddard Space Flight Center. We use a 4-month moving lag window to reduce the computational costs for projecting the flux perturbation ensemble into observation space long after (>4 months) their emissions, beyond which time it is difficult to distinguish between the emitted signal from variations in the ambient background atmosphere[15]. To calculate sequentially a posteriori estimate and the associated uncertainty via Eqs. (2) and (3), we use an efficient numerical LU solver[15].

Our a priori $CO_2$ flux inventory includes (1) monthly biomass burning emission[83]; (2) monthly fossil fuel emissions[84]; (3) monthly climatological ocean fluxes[85]; and (4) 3-hourly terrestrial biosphere fluxes[86]. Our $CO_2$ model calculations follow closely our previous calculations[38].

Our a priori $CH_4$ fluxes from nature include (1) monthly wetland emissions, including rice paddies[87]; (2) monthly fire $CH_4$ emissions[83]; and (3) termite emissions[88]. We include emissions from geological macroseeps[89,90]. For areal seepage, we use the sedimentary basins (microseepage) and potential geothermal seepage maps[89], with the emission factor taken from[91]. For a priori anthropogenic emissions we use the EDGAR v4.41 global emission inventory[92] that includes various sources related to human activities (e.g., oil and gas industry, coal mining, livestock, and waste). We use monthly 3D fields of the hydroxyl radical, consistent with observed values for the lifetime of methyl chloroform, from the GEOS-Chem $HO_x$–$NO_x$–$O_x$ chemistry simulation[93,94] to describe the main oxidation sink of $CH_4$[11]. Using fixed, the archived field of OH allows us to linearly decompose total $CH_4$ into contributions from individual sources and geographical regions. This approach greatly simplifies our gain matrix calculation (Eq. (3)).

For simplicity, we assume a fixed uncertainty of 40% for coefficients corresponding to a priori $CO_2$ fluxes over each sub-region, and a larger uncertainty (60%) for the corresponding $CH_4$ emissions. We also assume that a priori errors for the same gas are correlated with a spatial correlation length of 300 km and with a temporal correlation of 1 month. Our experiments show that our main results, e.g., the correlation between $CH_4$ flux and SST anomalies, are largely insensitive to either a higher (e.g., 20% higher) assumed a priori uncertainty or longer (e.g., 400 km) correlation length.

## Data availability

All the data and materials used in this study are freely available. The NOAA ObsPack data products (https://esrl.noaa.gov/gmd/ccgg/obspack/) and TCCON data (https://tccondata.org) are available subject to their fair use policies. The University of Leicester GOSAT Proxy v9.0 XCH4 data are available from the Centre for Environmental Data Analysis data repository at https://doi.org/10.5285/18ef8247f52a4cb6a14013f8235cc1eb. CHIRPS data are available at https://www.chc.ucsb.edu/data; NOAA OISST v2 data are available at https://psl.noaa.gov/; and the GRACE datasets are available at http://grace.jpl.nasa.gov.

## Code availability

The community-led GEOS-Chem model of atmospheric chemistry and model is maintained centrally by Harvard University (http://geos-chem.seas.harvard.edu), and is available on request. The ensemble Kalman filter code is publicly available as PyOSSE (https://www.nceo.ac.uk/data-tools/atmospheric-tools/).

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

## Acknowledgements

We thank all the scientists that submitted data to the $CO_2$ and $CH_4$ Observation Package (ObsPack) data products, coordinated by NOAA ESRL, and making them freely available for carbon cycle research. Similarly, we thank the individual investigators who collect $XCO_2$ and $XCH_4$ data as part of the Total Carbon Column Observing Network (TCCON). We thank the Japanese Aerospace Exploration Agency, National Institute for Environmental Studies, and the Ministry of Environment for the GOSAT data and their continued support as part of the Joint Research Agreements at the Universities of Edinburgh and Leicester. GOSAT retrievals were processed using the ALICE High-Performance Computing Facility at the University of Leicester. We thank Hartmut Bösch for useful discussions about the GOSAT data products. We also thank the GEOS-Chem community, particularly the team at Harvard University who help maintain the GEOS-Chem model, and the NASA Global Modeling and Assimilation Office (GMAO) who provide the MERRA2 data product. We gratefully acknowledge the GRACE data products that are supported by the NASA MEaSUREs program. The CMAP rainfall dataset and the NOAA OISST v2 data are provided by the NOAA/OAR/ESRL PSL, Boulder, Colorado, USA. L.F., P.I.P., and R.J.P. acknowledge support from the UK National Centre for Earth Observation funded by the National Environment Research Council (NE/R016518/1); R.J.P. also acknowledges funding from grant NE/N018079/1.

## Author contributions

L.F. and P.I.P. originated the ideas and designed the experiments, P.I.P. and L.F. led the writing of the paper with contributions from the coauthors S.Z., R.J.P., and Y.L.

## Competing interests

The authors declare no competing interests.
