## [Peer Review File · Nature Communications]

Title: Tropical methane emissions explain large fraction of recent changes in global atmospheric methane growth rateREVIEWER COMMENTS

Reviewer #1 (Remarks to the Author):

Review of “The role of tropical climate variations in the global atmospheric growth of methane”

The paper quantifies regional changes in the release of methane, using model and satellite outputs, and identifies a few tropical areas as being key in the recent increase in the release of methane. It is found that this increase is driven by changes in rainfall, driven by SSTs. This link between tropical SSTs and tropical methane emissions may offer some degree of potential predictability for future methane increases.

The paper is very important and the results are of substantial significance to this field, and other fields as it concerns the release of methane, a powerful greenhouse gas. It also has important implications for tropical meteorologists to consider the impacts of rainfall changes on methane emissions.

The methodology is well explained and clear. There are some (minor) revisions required in order to make the paper suitable for publication. My major concerns are the use of reanalysis precipitation across the tropics, rather than an observation based precipitation dataset and the short length of the record. Furthermore, some of the calculations lack statistical significance testing.

1) It is stated that both MERRA2 precipitation and CHIRPS precipitation are used. However, it is not clear when each of the precipitation datasets is used, although in Figure 3 it is stated that MERRA precipitation is used. I would strongly question the use of a reanalysis precipitation dataset over the tropics; a number of studies have indicated that they are not reliable. If it is to be used some evaluation against satellite/gauge-based rainfall datasets should be conducted and included in the Supplementary Information.

If the motivation for using reanalysis precipitation arose from the need to have precipitation estimates over the ocean then I would advise the use of GPCP, IMERG GPM or TRMM. It is also not stated whether daily or monthly rainfall is used.

2) I appreciate much of the data is only available over the 2010-19 period, however, this is a very short time period for calculating trends and correlations. While extending this analysis over a longer period may not be possible, correlations between SSTs, precipitation and possibly wetland extent/ lake levels could be extended over a longer period. This would add useful information to the conclusions, that we expect methane to be correlated with SST over longer timescales.

3) Figure 2a shows the 2015 minus 2010 change. It is stated in the text that this plot shows where methane emissions have changed over the study period – yet this plot only shows the change between 2 years. Another 2 years could be chosen that would show an opposite signal. I would suggest replacing this figure with a figure showing the trend over the entire study period. Is any statistical significance computed for these trends? I appreciate that it may be weak on such a short timeseries, but may be

interesting to explore. This also applies to the other trends in Figure S4.

4) Figure 2b includes the mean values as dashed line – I think it would be more useful if these lines showed the trend. The reader can easily discern that that methane emissions are higher over South America than India, but the trend is harder to discern by eye.

5) Line 85 - “Emissions from mainland and maritime tropical Southeast Asia significantly depart from the 2010-2019 mean only during 2015” – what about 2016-18 where large negative anomalies are seen? Also, on Figure 2c please align the x-axis tick marks with the bars to make it easier to read.

6) On Figure S4 please change the colour bar so that more rainfall is blue – I find this much more intuitive. Also, please provide some additional justification of why these regions were chosen. The largest changes over Africa look to be to the south of the region chosen, and the largest changes over South America look to be slightly to the east of the region chosen.

7) On Figure 3 it is stated that MERRA2 SST is used yet earlier it is stated that NOAA SST is used. I would imagine NOAA SST is the more reliable product, thus why has MERRA2 SST been used? Also, see my earlier comment about the use of MERRA2 precipitation over the tropics. Please add statistical significance to the maps.

8) When computing the correlations in Figure 3 is the same analysis repeated but with the seasonal cycle removed? I feel that it is important to be able to capture the interannual variability in methane emissions, but this linear and cubic fit may just be doing a good job of capturing the seasonal cycle and not the interannual variability. Please repeat this analysis with the seasonal cycle removed.

9) Where is the water table height used? It is mentioned as a dataset that is utilised, yet not mentioned in any of the plots in the main paper. Are there any correlations between methane and water table height computed? This would be an interesting addition, especially if there are differences between the methane/rainfall and methane/water level correlations.

10) Line 101 - “Conversely, rainfall over the Southwest has remained consistently higher than climatological values over our study period” – over what period was your climatology calculated? This is not clearly stated. In the line above it says climatology is 2010-19 but rainfall can’t be higher than climatology over 2010-19 if the climatology is calculated over 2010-19?

11) Please expand the discussion of the East Africa/IOD correlation (line 122-128). It is stated that the IOD is used, but it is not mentioned that the SST pattern displayed in Figure 3c closely matches the IOD pattern. Please discuss this – in the paragraph above you explain the choice of the SST correlation better. Also, in Table 2 there are fairly strong correlations between the IOD and MAM methane whereas few studies have found a strong correlation between MAM rainfall and the IOD. Please comment on this in the text with suitable references. Why are the dry seasons considered for South America but not for East Africa?

12) The main conclusion of the paper is “Our calculations show that the SST anomalies over tropical oceans have skill in describing variations in a posteriori CH₄ flux variations over tropical South America and tropical East Africa, via changes in rainfall”. While the results of this study do support this, this conclusion is based on a few correlation values. This could be strengthened e.g. by producing maps similar to Figure 3 but using methane emissions instead of rainfall and indicating regions where the correlation is statistically significant. I feel this would strengthen the conclusions. Or using SST in the linear/cubic fit models rather than just temperature and rainfall.

Reviewer #2 (Remarks to the Author):

The manuscript "The role of tropical climate variations in the global atmospheric growth of methane" by Feng et al investigates changes in the global methane budget over the past decade. In general, I found the paper to be well written and thorough, with important potential implications. There are a few major and minor suggestions I have, as outlined below (hopefully in a coherent fashion, they touch various aspects):

Major points

A) Figure 3: If I look at the right column and the correlation coefficients, I have a hard time to figure out whether most of the correlation is driven just by the mean seasonal cycle or not. To really make a robust and valid statement, the authors should try to separate the impact on de-seasonalized time series to decouple the correlations from the mean seasonal cycle and answer the following questions:

1) Does the regression model explain shift in the seasonal cycle from its climatology?

2) When aggregated to annual fluxes, can this regression model explain the IAV in fluxes and trends?

Visually, I would assume this might be true for 3a but not necessarily 3c, especially if you omit the last year.

I think this is an important point to thoroughly double-check as you really need to separate out correlations driven by the seasonal climatology.

B) Table 1:

It surprised me to see that the posterior-prior flux differences approach more than 2sigma of the prior uncertainty towards the end of the time-period. It almost looks as if the priors were a bit too tight here. To better understand the variations between African, tr. American and Indian IAV changes, it would be prudent to provide these regions separately in the table as well (with prior and posterior info). It would be important to see whether all regions pushed against their 2sigma prior uncertainty. This is just to make sure that all these regions had a roughly similar total prior uncertainty.

Also, what is driving your prior changes in wetland emissions? I couldn't find a correlation between the prior and posterior at all but think the analysis might be more robust if you just used a climatology as prior, not a wetland prior with IAV.

C) Proxy inversions:

I applaud the use of a direct inversion of the proxy. One minor problem could be that tropical CO₂ fluxes are not that greatly constrained by the surface network. For the tropical regions you outline, are you sure that none of the CH₄ fluxes are affected by changes in your CO₂ inversion scheme? In actuality, there might be co-variations between the fluxes even in reality. It would be good to also show changes in your derived tropical CO₂ fluxes (SOM is fine) to ensure that these are not driving your CH₄ fluxes (i.e. compensating each other).

Minor points:

Figure 2a: For me, it is hard to discern small changes on that colorbar. Can you use fewer bins? Between 3-5, I can't differentiate (units missing as well).

Figure 2c: Can you add the prior and prior uncertainties aggregated by region here as well?

Line 133: I think this statement can only be made if you find a clear correlation of SST variations and annual averages of methane emissions. See major comment above

Lines 83-84: This is a key and important message. Please just outline prior uncertainties for these regions as well. We need to ensure that these are robust and not affected by the constraints.

Lines 60+: The global mean emissions are not necessarily related to growth rates (the net fluxes are). As stated, it is a bit misleading

Line 29: "Wetlands" is sandwiched in between even though it is the main driver. Maybe order sources by importance.

Line 31: Stratospheric sink (not through OH!) should also be mentioned if you want to be complete.

Reviewer #3 (Remarks to the Author):

Feng et al., in their manuscript perform an inverse modelling study for the period 2010-2019 of atmospheric methane based on observations from the instrument on board the GOSAT satellite. The focus of their investigation is the tropical region, a region from which wetland emissions are known to be a dominant source of methane emissions. The authors conclude that the interannual variability in methane emissions from this region is strongly influenced by the impact of temperature and rainfall on methane emissions from wetlands, with these effects highly correlated with SST (Sea Surface Temperatures) over the tropical oceans. The authors highlight that SST variations can be well predicted, and that therefore the links between SST variability and variability in methane emissions from tropical wetlands could be used to predict variability in the growth rate of atmospheric methane. The manuscript is clearly written, the authors describe their methodology and data sources well, and the

chosen methods and data are appropriate for the investigation. This reviewer however finds this overall conclusion of the manuscript to be generally unremarkable, and not of sufficient significance or originality for a highlight paper. The strong influence of meteorological variability on emissions of methane from wetlands is discussed thoroughly in the recent review by Saunio et al. (2020), which is cited as reference 19 in the current manuscript, as well as several of the references therein. The influence of tropical SSTs on the variability of temperature and precipitation in the tropics is also well-established (eg. through numerous studies on the well-known El Niño- Southern Oscillation phenomenon). A strong predictive link between tropical SSTs and the variability of the atmospheric methane growth rate is therefore completely to be expected.

The long-term growth rate of atmospheric methane is in fact a topic of interest to a wide readership, since methane is a potent greenhouse gas, and there is currently no consensus on the cause of the long-term trends in atmospheric methane. Feng et al. allude to this briefly through their citation of the work of Turner et al. (2019) on line 37, but the competing explanations for the growth of atmospheric methane are not discussed by Feng et al. in their manuscript, and by focusing on the short-term variability of atmospheric methane, they make no contribution to understanding the causes of the long-term trends.

In summary, while the work submitted by Feng et al. is well described and methodologically sound, I do not find it sufficiently original or significant for publication in Nature Communications and recommend rejection of the manuscript.

We are extremely grateful for the reviewers' helpful comments. We apologise for the delay in our responses – we decided to test the robustness of our results against model resolution (we now report results at a higher resolution) and we also suffered from unavoidable delays associated with Covid-19. We have answered the reviewer comments point by point below. Also corresponding modifications have been made in the manuscript.

Reviewer #1 (Remarks to the Author):

Review of “The role of tropical climate variations in the global atmospheric growth of methane”

The paper quantifies regional changes in the release of methane, using model and satellite outputs, and identifies a few tropical areas as being key in the recent increase in the release of methane. It is found that this increase is driven by changes in rainfall, driven by SSTs. This link between tropical SSTs and tropical methane emissions may offer some degree of potential predictability for future methane increases.

The paper is very important and the results are of substantial significance to this field, 21w3and other fields as it concerns the release of methane, a powerful greenhouse gas. It also has important implications for tropical meteorologists to consider the impacts of rainfall changes on methane emissions.

The methodology is well explained and clear. There are some (minor) revisions required in order to make the paper suitable for publication. My major concerns are the use of reanalysis precipitation across the tropics, rather than an observation based precipitation dataset and the short length of the record. Furthermore, some of the calculations lack statistical significance testing.

1) It is stated that both MERRA2 precipitation and CHIRPS precipitation are used. However, it is not clear when each of the precipitation datasets is used, although in Figure 3 it is stated that MERRA precipitation is used. I would strongly question the use of a reanalysis precipitation dataset over the tropics; a number of studies have indicated that they are not reliable. If it is to be used some evaluation against satellite/gauge-based rainfall datasets should be conducted and included in the Supplementary Information.

If the motivation for using reanalysis precipitation arose from the need to have precipitation estimates over the ocean then I would advise the use of GPCP, IMERG GPM or TRMM. It is also not stated whether daily or monthly rainfall is used.

We fully agreed that there are large uncertainties associated with precipitation datasets, including those that rely on model reanalysis and satellite/gauge-based measurements (Chen et al., 2019). Following the suggestions, we now use a third-party (CMAP) monthly precipitation data in our analysis, and we have accordingly revised the text and Figures 3, S4 and S5. Using this additional data does not change the main conclusions of our study.

2) I appreciate much of the data is only available over the 2010-19 period, however, this is a very short time period for calculating trends and correlations. While extending this analysis over a longer period may not be possible, correlations between SSTs,

precipitation and possibly wetland extent/ lake levels could be extended over a longer period. This would add useful information to the conclusions, that we expect methane to be correlated with SST over longer timescales.

This is a good point. The statistically significant correlations we find between SST anomalies and CH₄ emission anomalies reflect physical processes associated changes in continental rainfall and surface temperature. This can only be further tested by extending the satellite CH₄ record, which will be possible with upcoming missions; the *in situ* record is too sparse over the tropics.

Following this reviewer suggest we now report in Figure S5 correlations between rainfall and the NINO3.4 index, the Pacific-Atlantic Dipole index (as discussed in the main paper) and the IOD from 2000 to 2019. This analysis builds on other recent studies, e.g., Wainwright et al, 2019.

A challenge for such studies is the inconsistency between the available precipitation data sets. We show below an example of the correlation between the IOD SST index and rainfall anomalies over Africa during the short rains (OND), 2005-2019. Both the TAMSAT v3.1 and GPCP v2.3 rainfall datasets show strong (positive) correlation over eastern Africa (East Africa and the Horn of Africa). However, we do not find a similar correlation using CHIRPS v2 rainfall data. Using the GRACE LWE is more difficult and will not necessarily reflect where the rain falls but where the ground water ends up, driven by basin dynamics. Comparing and contrasting the value of using different datasets is beyond the scope of this paper.

Figure A1: Pearson correlation between IOD SST anomaly and the African OND rainfall anomaly based on the GPCP, TAMSAT and CHIRPS dataset during 2005-2019. For comparison the correlation with GRACE LWE and IOD SST anomaly is also shown.

3) *Figure 2a shows the 2015 minus 2010 change. It is stated in the text that this plot shows where methane emissions have changed over the study period – yet this plot only shows the change between 2 years. Another 2 years could be chosen that would show an opposite signal. I would suggest replacing this figure with a figure showing the trend over the entire study period. Is any statistical significance computed for these trends? I appreciate that it may be weak on such a short timeseries, but may be interesting to explore. This also applies to the other trends in Figure S4.*

Thanks for the suggestion. we have now replaced Figure 2a with differences between two 5-year means for 2010-2014 and 2015-2019, respectively. We now also highlight in Figure 3 and Figure S5 significant p-values of the presented Pearson correlations. In Figure S4 we only show data that have a trend significantly different from zero ($p < 0.05$).

5) *Line 85 – “Emissions from mainland and maritime tropical Southeast Asia significantly depart from the 2010-2019 mean only during 2015” – what about 2016-18 where large negative anomalies are seen? Also, on Figure 2c please align the x-axis tick marks with the bars to make it easier to read. Emissions from mainland and maritime tropical Southeast Asia significantly depart from the 2010-2019 mean only during 2015” – what about 2016-18 where large*

Point well taken. We have changed the plot, and revised the corresponding text:

‘Emissions from mainland and maritime tropical Southeast Asia have significantly decreased from 2010 to 2019, driven primarily by reduced rainfall...’

6) *On Figure S4 please change the colour bar so that more rainfall is blue – I find this much more intuitive. Also, please provide some additional justification of why these regions were chosen. The largest changes over Africa look to be to the south of the region chosen, and the largest changes over South America look to be slightly to the east of the region chosen.*

We have changed the colour bar in Figure S4 following the suggestion. We have also extended the definition of the three regions to cover larger areas.

7) *On Figure 3 it is stated that MERRA2 SST is used yet earlier it is stated that NOAA SST is used. I would imagine NOAA SST is the more reliable product, thus why has MERRA2 SST been used? Also, see my earlier comment about the use of MERRA2 precipitation over the tropics. Please add statistical significance to the maps.*

We now use NOAA OI SST data and CMAP rainfall data in the revision.

We now highlight regions the p values associated with the Pearson correlations (Figure 3, Figure S5).

8) *When computing the correlations in Figure 3 is the same analysis repeated but with the seasonal cycle removed? I feel that it is important to be able to capture the interannual variability in methane emissions, but this linear and cubic fit may just be doing a good job of capturing the seasonal cycle and not the interannual variability. Please repeat this analysis with the seasonal cycle removed.*

This was an oversight on our part for which we apologise. We have revised Figure 3 and subsequent analysis to consider how CH₄ emission anomalies (de-seasonalised emission time series) can be described by anomalies in temperature, rainfall, and soil moisture using linear and quadratic regression models.

9) *Where is the water table height used? It is mentioned as a dataset that is utilised, yet not mentioned in any of the plots in the main paper. Are there any correlations between methane and water table height computed? This would be an interesting addition, especially if there are differences between the methane/rainfall and methane/water level correlations.*

We have used GRACE LWE data in Figure S4, also we have added a sentence in the main text:

‘These contrasting signals are reflected in the opposite trends for GRACE LWE (Figure S4) ‘

As stated above, we now include MERRA2 soil moisture instead of GRACE LWE in the linear and quadratic model to fit the CH₄ flux anomaly, because GRACE data does not cover the full study period.

10) *Line 101 - “Conversely, rainfall over the Southwest has remained consistently higher than climatological values over our study period” – over what period was your climatology calculated? This is not clearly stated. In the line above it says climatology is 2010-19 but rainfall can’t be higher than climatology over 2010-19 if the climatology is calculated over 2010-19?*

We have clarified in the manuscript that the climatology covers 2000-2019.

11) *Please expand the discussion of the East Africa/IOD correlation (line 122-128). It is stated that the IOD is used, but it is not mentioned that the SST pattern displayed in Figure 3c closely matches the IOD pattern. Please discuss this – in the paragraph above you explain the choice of the SST correlation better. Also, in Table 2 there are fairly strong correlations between the IOD and MAM methane whereas few studies have found a strong correlation between MAM rainfall and the IOD. Please comment on this in the text with suitable references. Why are the dry seasons considered for South America but not for East Africa?*

In the main text, to help clarify this discussion we have added:

‘Variations in the short rains determined by the CMAP dataset show a strong positive correlation with SST anomalies over the western Indian Ocean, and a strong negative correlation with SST anomalies over the eastern Indian Ocean, consistent with previous studies using different rainfall datasets (e.g., Wainwright et al., 2019)’

For the higher-resolution transport model we have now used for this study, we only find moderate correlations between IOD SST and CH₄ flux anomalies during MAM. We find that during MAM, CMAP rainfall, MERRA2 temperature, and GRACE LWE show similar correlations with IOD SST, and with NINO3.4/PAD SST anomalies.

It is difficult to separate the impacts of the 2015/2016 El Niño event with those driven by IOD SST anomalies. Consequently, we cannot draw a definitive conclusion about the moderate correlation between the IOD SST anomaly and CH₄ flux anomaly for MAM.

Over eastern Africa, CH₄ emissions from wetlands associated with the long rains (MAM) and short rains (OND). Historically, the long rains were responsible for sustaining agriculture over the regions but the relative importance of these two rain seasons is already beginning to change (e.g., Lunt et al, 2021), with climate model ensemble projecting that the short rain season will soon dominate the region.

12) The main conclusion of the paper is "Our calculations show that the SST anomalies over tropical oceans have skill in describing variations in a posteriori CH₄ flux variations over tropical South America and tropical East Africa, via changes in rainfall". While the results of this study do support this, this conclusion is based on a few correlation values. This could be strengthened e.g., by producing maps similar to Figure 3 but using methane emissions instead of rainfall and indicating regions where the correlation is statistically significant. I feel this would strengthen the conclusions. Or using SST in the linear/cubic fit models rather than just temperature and rainfall.

Wetland emissions of CH₄ are not exclusively determined by immediate changes in rainfalls. They are also driven by changes in surface temperature, water table depth, and the availability of substrate, which all act on different timescales. These environmental variables are all affected, but not solely determined, by SST anomalies. Our work suggests that correlations between CH₄ flux anomalies and NINO3.4/PAD/IOD SST anomalies vary between seasons. Consequently, we do not expect an aseasonal relationship between SST variations and CH₄ flux anomalies.

Reviewer #2 (Remarks to the Author):

The manuscript "The role of tropical climate variations in the global atmospheric growth of methane" by Feng et al investigates changes in the global methane budget over the past decade. In general, I found the paper to be well written and thorough, with important potential implications. There are a few major and minor suggestions I have, as outlined below (hopefully in a coherent fashion, they touch various aspects):

Major points

A) Figure 3: If I look at the right column and the correlation coefficients, I have a hard time to figure out whether most of the correlation is driven just by the mean seasonal cycle or not. To really make a robust and valid statement, the authors should try to separate the impact on de-seasonalized time series to decouple the correlations from the mean seasonal cycle and answer the following questions:

Great suggestion, also raised by reviewer 1. We have revised Figure 3 and subsequent analysis to consider how CH₄ emission anomalies (de-seasonalised emission time series) can be described by anomalies in temperature, rainfall, and soil moisture using linear and quadratic regression models.

1) Does the regression model explain shift in the seasonal cycle from its climatology?

As shown in the new Figure 3, the correlation coefficients associated with the regression models that use the anomalies of driving data (e.g., temperature and rainfall) now explain 40-60% of the year-to-year variations in CH₄ emissions.

2) When aggregated to annual fluxes, can this regression model explain the IAV in fluxes and trends? Visually, I would assume this might be true for 3a but not necessarily 3c, especially if you omit the last year. I think this is an important point to thoroughly double-check as you really need to separate out correlations driven by the seasonal climatology.

See above answer.

B) Table 1:

It surprised me to see that the posterior-prior flux differences approach more than 2sigma of the prior uncertainty towards the end of the time-period. It almost looks as if the priors were a bit too tight here. To better understand the variations between African, tr. American and Indian IAV changes, it would be prudent to provide these regions separately in the table as well (with prior and posterior info). It would be important to see whether all regions pushed against their 2sigma prior uncertainty. This is just to make sure that all these regions had a roughly similar total prior uncertainty.

Posterior estimates inferred from top-down inversions can significantly differ from the corresponding prior estimate, even beyond 2-sigma of the prior uncertainty. It is partially due to the quality of the prior estimates and to the assumption about the prior error covariance.

In this study, we adopted a fixed wetland emission after 2016 so the resulting prior model CH₄ growth rate was systematically below values inferred by NOAA data. The data assimilation procedure, described by the ensemble Kalman filter, corrected this underestimation by increasing emissions mainly from tropical regions (Figure 1, Table 1 and the new Table S4 suggested by the reviewer). The resulting posterior emissions have much smaller uncertainties than prior values (typically reduced by over 50%). This indicates that the trends and the IAVs reported in our study are mainly determined by observations, which is further confirmed by sensitivity experiments that use higher or lower prior uncertainties. As described in the supplementary material, we already assume a large uncertainty (60%) for each subregion. However, when we aggregate subregions to determine continental-scale prior uncertainties we assume only random errors that will potentially underestimate the prior uncertainty. We have not considered large systematic errors associated with the inventory, e.g., tropical North African emissions after 2016 (Lunt et al, 2019, 2021).

2. Also, what is driving your prior changes in wetland emissions? I couldn't find a correlation between the prior and posterior at all but think the analysis might be more robust if you just used a climatology as prior, not a wetland prior with IAV.

In our study, the wetland emission is described by WetCHARTs v1.0 (Bloom et al., 2016). Wetland emissions are mainly driven by changes in groundwater, surface temperature and substrate availability, but in our calculations these emissions are fixed after 2016. Table S2, suggested by the reviewer, shows that the posterior fluxes have much larger interannual variations, with significantly reduced (by more than 50%) uncertainty. In the revised SI, we now emphasize that the IAV for regional posterior methane emission estimates are mainly determined by observations from GOSAT and the surface *in situ* network (Table S2).

I applaud the use of a direct inversion of the proxy. One minor problem could be that tropical CO₂ fluxes are not that greatly constrained by the surface network. For the tropical regions you outlines, are you sure that none of the CH₄ fluxes are affected by changes in your CO₂ inversion scheme? In actuality, there might be co-variations between the fluxes even in reality. It would be good to also show changes in your derived tropical CO₂ fluxes (SOM is fine) to ensure that these are not driving your CH₄ fluxes (i.e. compensating each other).

To clarify, we use *in situ* CH₄ and CO₂ observations as large-scale anchors for the information provided by the X_{CH₄}:X_{CO₂} ratios provided by GOSAT. We agree with this reviewer and acknowledge that the current *in situ* network coverage is not ideal, with gaps over the tropics.

To address this point raised by the reviewer, we ran two inversion runs that share the same basic configuration but with a 15% smaller prior uncertainty than the one reported in main text. The control run used GOSAT X_{CH₄}:X_{CO₂} ratio data and the surface CH₄ and CO₂ observations by the NOAA surface network. The sensitivity run used the control run data but also GOSAT full-physics X_{CO₂} retrievals (provided by the NASA ACOS team) as extra constraints on model CO₂. We find that including the GOSAT full-physics X_{CO₂} data increases net CO₂ emissions over the terrestrial tropics (e.g., Palmer et al, 2019) and consequently results in higher CH₄ emissions, particularly over northeast tropical Africa (Figure A2). We find the magnitude and spatial distribution of the correlations between CH₄ emission anomalies and SST anomaly for the sensitivity and the CH₄ emission trends are similar to those from the control run (Figure A3).

The correlation between IOD SST anomalies and CH₄ emission anomalies during the short rains (OND) is 0.54 for the sensitivity run and 0.55 for the control run. The correlation between the PAD SST anomalies and CH₄ emission anomalies over North Tropical America during wet seasons (dry seasons) is -0.69 (-0.59) for the sensitivity run compared to -0.68 (-0.61) for the control run. Over the Southwest Amazon, including GOSAT full physics X_{CO₂} retrievals slightly increases the correlation for the dry (wet) season from 0.71 (0.75) for the control run to 0.76 (0.76) for the sensitivity run.

Figure A2: Time series of CH₄ emission from the North Tropical America, Southwest Amazon and Northeast Tropical Africa defined In Figure 3. The black lines represent the control inversion using the proxy GOSAT X_{CH₄}:X_{CO₂} ratio retrievals together with the surface CO₂ and CH₄ observations by the NOAA insitu network. The red lines correspond to the sensitivity experiment with GOSAT Full-physics XCO₂ retrievals as extra constraints.

Figure A3: Trends of posterior CH₄ flux estimated in the control run (top), and in the experiment run with extra constraints by ACOS GOSAT XCO₂ data (bottom).

Minor points:

Figure 2a: For me, it is hard to discern small changes on that colorbar. Can you use fewer bins? Between 3-5, I can't differentiate (units missing as well).

We have now revised Figure 2 to display the difference between two 5-year periods.

Figure 2c: Can you add the prior and prior uncertainties aggregated by region here as well?

Line 133: I think this statement can only be made if you find a clear correlation of SST variations and annual averages of methane emissions. See major comment above

To address this point we have changed Figure 2 to highlight deviations from their 10-year mean. We have also added Table S2 to report the prior and posterior annual values following this reviewer suggestion. The table confirms that IAVs of tropical regions are mainly determined by observations.

Lines 83-84: This is a key and important message. Please just outline prior uncertainties for these regions as well. We need to ensure that these are robust and not affected by the constraints.

To address this point we have added Table S2 to report the prior and posterior fluxes and their uncertainties.

Lines 60+: The global mean emissions are not necessarily related to growth rates (the net fluxes are). As stated, it is a bit misleading

We change to state they are net emissions

Line 29: "Wetlands" is sandwiched in between even though it is the main driver. Maybe order sources by importance.

Agreed. We have now changed the order of source list following the suggestion as below:

Emissions are from wetlands, agriculture (e.g. rice paddies, ruminants and waste), anthropogenic emissions (e.g. fossil fuel production and consumption), burning of biomass and biofuels}, with minor emissions from, for example, geological seepage, termites, inland water, and oceans (Saunois et al, 2020)

Line 31: Stratospheric sink (not through OH!) should also be mentioned if you want to be complete.

Thanks for spotting this error. We have now added the following statement.

'Reaction with chlorine atoms also plays a role in determining the lifetime of atmospheric methane.'

Reviewer #3 (Remarks to the Author):

Feng et al., in their manuscript perform an inverse modelling study for the period 2010-2019 of atmospheric methane based on observations from the instrument on board the GOSAT satellite. The focus of their investigation is the tropical region, a region from which wetland emissions are known to be a dominant source of methane emissions. The authors conclude that the interannual variability in methane emissions from this region is strongly influenced by the impact of temperature and rainfall on methane emissions from wetlands, with these effects highly correlated with SST (Sea Surface Temperatures) over the tropical oceans. The authors highlight that SST variations can be well predicted, and that therefore the links between SST variability and variability in methane emissions from tropical wetlands could be used to predict variability in the growth rate of atmospheric methane. The manuscript is clearly written, the authors describe their methodology and data sources well, and the chosen methods and data are appropriate for the investigation. This reviewer however finds this overall conclusion of the manuscript to be generally unremarkable, and not of sufficient significance or originality for a highlight paper. The strong influence of meteorological variability on emissions of methane from wetlands is discussed thoroughly in the recent review by Saunois et al. (2020), which is cited as reference 19 in the current manuscript, as well as several of the references therein. The influence of tropical SSTs on the variability of temperature and precipitation in the tropics is also well-established (eg. through numerous studies on the well-known El Nino- Southern Oscillation phenomenon). A strong predictive link between tropical SSTs and the variability of the atmospheric methane growth rate is therefore completely to be expected. The long-term growth rate of atmospheric methane is in fact a topic of interest to a wide readership, since methane is a potent greenhouse gas, and there is currently no consensus on the cause of the long-term trends in atmospheric methane. Feng et al. allude to this briefly through their citation of the work of Turner et al. (2019) on line 37, but the competing explanations for the growth of atmospheric methane are not discussed by Feng et al. in their manuscript, and by focusing on the short-term variability of atmospheric methane, they make no contribution to understanding the causes of the long-term trends.

In summary, while the work submitted by Feng et al. is well described and methodologically sound, I do not find it sufficiently original or significant for publication in Nature Communications and recommend rejection of the manuscript.

We thank this reviewer for their summary of our manuscript but respectfully disagree with their final assessment. We find that large-scale climate variations play a key role in determining variations in CH₄ emission, particularly relevant to tropical South America and tropical Africa.

References

Lunt, M. F., Palmer, P. I., Feng, L., Taylor, C. M., Boesch, H., and Parker, R. J.: An increase in methane emissions from tropical Africa between 2010 and 2016 inferred from satellite data, *Atmos. Chem. Phys.*, 19, 14721–14740, <https://doi.org/10.5194/acp-19-14721-2019>, 2019.

Lunt, M. F., Palmer, P. I., Lorente, A., Borsdorff, T., Landgraf, J., Parker, R. J., and Boesch, H. Rain-fed pulses of methane from East Africa during 2018–2019 contributed to atmospheric growth rate. *Environmental Research Letters*, 16(2):024021, feb 2021.

Palmer, P. I., L. Feng, D. Baker, F. Chevallier, H. Boesch, P. Somkuti, “Net carbon emissions from African land biosphere dominate pan-tropical atmospheric CO₂ signal”, *Nature Comm.*, <https://doi.org/10.1038/s41467-019-11097-w>, 2019.

Wainwright, C.M., Marsham, J.H., Keane, R.J., Rowell, D.P., Finney, D.L., Black, E. and Allan, R.P. (2019): ‘Eastern African Paradox’ rainfall decline due to shorter not less intense Long Rains. *npj Climate and Atmospheric Science*, 2(1), 1–9, <https://doi.org/10.1038/s41612-019-0091-7>.

REVIEWERS' COMMENTS

Reviewer #1 (Remarks to the Author):

Review of “The role of tropical climate variations in the global atmospheric growth of methane”

This paper uses models and satellite observations to calculate regional emissions estimates of methane. Tropical terrestrial regions that contribute more to global methane emissions, and contribute more to the variability in global methane emissions are highlighted. The emissions over some regions are then linked to variability in temperature and rainfall over the period 2010-2019, and to SST variability. This link between tropical SSTs and tropical methane emissions may offer some degree of potential predictability for future methane increases.

The paper is important, as methane is an important greenhouse gas. The results in the paper indicate that tropical SST patterns may offer some degree of predictability for methane emissions, which may be useful in managing future emissions.

The authors have addressed many of my concerns, and I am happy to recommend publication, following a few minor revisions.

Minor comments:

- 1) Line 57 – why is NOAA OISST not named here? I found it slightly annoying to have to go to the supplementary information to find this.
- 2) On the figures subpanels (a,b,c,) are not labelled – please add these to aid readability.
- 3) Line 88 – “emissions from mainland and maritime Tropical South East Asia”. I assume by this that you mean mainland and islands. However, I wonder if this could be interpreted as “land and ocean” which is not what is meant?
- 4) Line 90 – do you have a reference for the fire emissions part?
- 5) Line 107 – what is meant by “opposite trends in GRACE”? It seems to me from Figure S4 that where rainfall is increasing GRACE LWE is increasing? Do you mean opposite in the different regions?
- 6) In line 117 you refer to Table S2 as showing links with rainfall, temperature and soil moisture. Do you mean Table S3, and soil moisture is not included?
- 7) Line 117-8: you state that temperature and rainfall play comparable roles in driving methane, yet the correlation with rainfall is much stronger?

8) Line 119 – Do you mean table S3 again?

9) Figure S5 is not mentioned in the main manuscript. Please include reference to this.

Reviewer #2 (Remarks to the Author):

The authors did a good job in responding to the reviewer comments. I just have a few nit-picky comments:

Stratospheric CH₄ sink: What I meant is not really the same as the Chlorine sink (which is more related to the MBL). So maybe just mention OH sink as well as stratospheric losses as well as soil sinks. Right now, it sounds awkward to mention Chlorine only.

Figure 1: In A (labels in sub-plots also missing): The x-labels get awfully close together, just having numbers for 0,5,10,15,20 should suffice). Maybe a different colorscale (like viridis) would allow the readers to discern more subtle changes across regions.

I also respectfully disagree with reviewer #3. While I see part of the argument that they put forward in terms of originality, the authors of this paper not only see short-term IAVs but also an overall trend in tropical methane e

Responses to reviewer comments of NCOMMS-20-45031A

Below we provide a point-by-point response to the reviewer comments (red). We thank both reviewers for their additional comments and request for clarification.

Reviewer #1 (Remarks to the Author):

Review of “The role of tropical climate variations in the global atmospheric growth of methane”

This paper uses models and satellite observations to calculate regional emissions estimates of methane. Tropical terrestrial regions that contribute more to global methane emissions, and contribute more to the variability in global methane emissions are highlighted. The emissions over some regions are then linked to variability in temperature and rainfall over the period 2010-2019, and to SST variability. This link between tropical SSTs and tropical methane emissions may offer some degree of potential predictability for future methane increases.

The paper is important, as methane is an important greenhouse gas. The results in the paper indicate that tropical SST patterns may offer some degree of predictability for methane emissions, which may be useful in managing future emissions.

The authors have addressed many of my concerns, and I am happy to recommend publication, following a few minor revisions.

Minor comments:

1) Line 57 – why is NOAA OISST not named here? I found it slightly annoying to have to go to the supplementary information to find this.

Good point. We have now addressed this point.

2) On the figures subpanels (a,b,c,) are not labelled – please add these to aid readability.

We addressed this comment but have since removed them following the Nature formatting guidelines.

3) Line 88 – “emissions from mainland and maritime Tropical South East Asia”. I assume by this that you mean mainland and islands. However, I wonder if this could be interpreted as “land and ocean” which is not what is meant?

These are the geographical names for the two contrasting regions but we have stressed that maritime SE Asia includes the Island nations when we first use this term.

4) Line 90 – do you have a reference for the fire emissions part?

We have added the reference that describes our fire emission inventory.

5) Line 107 – what is meant by “opposite trends in GRACE”? It seems to me from Figure S4 that where rainfall is increasing GRACE LWE is increasing? Do you mean opposite in the different regions?

Yes, we have now corrected this statement.

6) In line 117 you refer to Table S2 as showing links with rainfall, temperature and soil moisture. Do you mean Table S3, and soil moisture is not included?

That was a typo so thanks for spotting it. It should, indeed, read Table S3. We have added the soil moisture column to Table S3.

7) Line 117-8: you state that temperature and rainfall play comparable roles in driving methane, yet the correlation with rainfall is much stronger?

What we mean is that temperature anomalies are also important but they are smaller and the statement has now been corrected.

8) Line 119 – Do you mean table S3 again?

Yes, it's the same typo as before. Apologies.

9) Figure S5 is not mentioned in the main manuscript. Please include reference to this.

It is not mentioned in the main manuscript but forms part of the wider discussion in the SI. Consequently, we have left the manuscript as it is.

Reviewer #2 (Remarks to the Author):

The authors did a good job in responding to the reviewer comments. I just have a few nit-picky comments:

Stratospheric CH₄ sink: What I meant is not really the same as the Chlorine sink (which is more related to the MBL). So maybe just mention OH sink as well as stratospheric losses as well as soil sinks. Right now, it sounds awkward to mention Chlorine only.

Agreed. This has now been fixed.

Figure 1: In A (labels in sub-plots also missing): The x-labels get awfully close together, just having numbers for 0,5,10,15,20 should suffice). Maybe a different colorscale (like viridis) would allow the readers to discern more subtle changes across regions.

Agreed. This has now been fixed.

I also respectfully disagree with reviewer #3. While I see part of the argument that they put forward in terms of originality, the authors of this paper not only see short-term IAVs but also an overall trend in tropical methane emissions, which can explain most of the global trends. If true, this is indeed important as typical methane wetland emissions inventories have little to no trend (see Turner et al), which is often why most long-term increases are only attributed to anthropogenic emissions, for which there is a trend in the prior.

Agreed.